# Cohort profile: The Social media, smartphone use and Self-harm in Young People (3S-YP) study–A prospective, observational cohort study of young people in contact with mental health services

**Amanda Bye** [1,2] *, **Ben Carter** [3], **Daniel Leightley** [1,4,5], **Kylee Trevillion** [6], **Maria Liakata** [7,8,9], **Stella Branthonne-Foster** [10], **Samantha Cross** [3], **Zohra Zenasni** [3], **Ewan Carr** [3], **Grace Williamson** [1,4], **Alba Vega Viyuela** [11,12], **Rina Dutta** [1,13]

1 Department of Psychological Medicine, Institute of Psychiatry, Psychology and Neuroscience, King's College London, London, United Kingdom, 2 Department of Child and Adolescent Psychiatry, Institute of Psychiatry, Psychology and Neuroscience, King's College London, London, United Kingdom, 3 Department of Biostatistics and Health Informatics, Institute of Psychiatry, Psychology and Neuroscience, King's College London, London, United Kingdom, 4 Institute of Psychiatry, King's Centre for Military Health Research, Psychology and Neuroscience, King's College London, London, United Kingdom, 5 School of Life Course & Population Sciences, King's College London, London, United Kingdom, 6 Health Service and Population Research Department, Institute of Psychiatry, Psychology and Neuroscience, King's College London, London, United Kingdom, 7 School of Electronic Engineering & Computer Science, Queen Mary, University of London, London, United Kingdom, 8 The Alan Turing Institute, London, United Kingdom, 9 University of Warwick, Warwick, United Kingdom, 10 Senior Service User Consultant, London, United Kingdom, 11 National Institute for Health and Care Research (NIHR) Clinical Research Network (CRN) South London, London, United Kingdom, 12 Cardiology Research Department, Health Research Institute, Fundación Jiménez Díaz Hospital, Madrid, Spain, 13 South London and Maudsley NHS Foundation Trust, London, United Kingdom

* amanda.bye@kcl.ac.uk

## Abstract

### Objectives

The Social media, Smartphone use and Self-Harm (3S-YP) study is a prospective observational cohort study to investigate the mechanisms underpinning associations between social media and smartphone use and self-harm in a clinical youth sample. We present here a comprehensive description of the cohort from baseline data and an overview of data available from baseline and follow-up assessments.

### Methods

Young people aged 13–25 years were recruited from a mental health trust in England and followed up for 6 months. Self-report data was collected at baseline and monthly during follow-up and linked with electronic health records (EHR) and user-generated data.

**Data Availability Statement:** Due to conditions of participant consent, and as it contains potentially

sensitive and identifiable information, the data supporting this article cannot be openly shared. To request access, email research.data@kcl.ac.uk. A descriptive record can be found in the King's College London research data repository, KORDS, at DOI: 10.18742/25043606.

**Funding:** "This work was supported by the Medical Research Council and Medical Research Foundation (grant number MR/S020365/1). This work was also part supported by the National Institute for Health and Care Research (NIHR) Maudsley Biomedical Research Centre (BRC) and King's College London, and the NIHR Clinical Research Network (CRN) South London. RD was also funded by a Clinician Scientist Fellowship from the Health Foundation in partnership with the Academy of Medical Sciences and her work is supported by the National Institute for Health Research (NIHR) Biomedical Research Centre at South London and Maudsley NHS Foundation Trust and King's College London. BC is also supported by the Nuffield Trust. ML is also supported by the Engineering and Physical Sciences Research Council (grant number EP/V030302/1) and The Alan Turing Institute (grant number EP/N510129/1). AB and EC are also supported by the National Institute for Health and Care Research (NIHR) Maudsley Biomedical Research Centre (BRC) and King's College London. The views expressed are those of the author(s) and not necessarily those of the MRC, the MRF, the NHS, the NIHR or the Department of Health and Social Care. The funders had no role in study design, data collection and analysis, decision to publish, or preparation of the manuscript. For the purposes of open access, the author has applied a Creative Commons Attribution (CC BY) licence to any Accepted Author Manuscript version arising from this submission".

**Competing interests:** The authors have declared that no competing interests exist.

## Findings

A total of 362 young people enrolled and provided baseline questionnaire data. Most participants had a history of self-harm according to clinical (n = 295, 81.5%) and broader definitions (n = 296, 81.8%). At baseline, there were high levels of current moderate/severe anxiety (n = 244; 67.4%), depression (n = 255; 70.4%) and sleep disturbance (n = 171; 47.2%). Over half used social media and smartphones after midnight on weekdays (n = 197, 54.4%; n = 215, 59.4%) and weekends (n = 241, 66.6%; n = 263, 72.7%), and half met the cut-off for problematic smartphone use (n = 177; 48.9%). Of the cohort, we have questionnaire data at month 6 from 230 (63.5%), EHR data from 345 (95.3%), social media data from 110 (30.4%) and smartphone data from 48 (13.3%).

## Conclusion

The 3S-YP study is the first prospective study with a clinical youth sample, for whom to investigate the impact of digital technology on youth mental health using novel data linkages. Baseline findings indicate self-harm, anxiety, depression, sleep disturbance and digital technology overuse are prevalent among clinical youth. Future analyses will explore associations between outcomes and exposures over time and compare self-report with user-generated data in this cohort.

## Introduction

Self-harm is one of the strongest risk factors for suicide and occurs most frequently in those aged under 25 years [1,2]. Younger people are also much more likely to be digitally connected, with 94% of 16–17 year olds having a social media profile in the UK [3] and 99% of 16–24 year olds owning a smartphone [4]. Although there is marked variation by age, even children as young as 3 years of age use social media and rates of smartphone ownership tends to increase with the move to secondary school as children approach 11 years [5].

There has been considerable debate about whether social media and smartphone use are responsible for the steep rise in youth mental health issues including self-harming, especially amongst teenage girls [6]. The evidence is mainly limited to surveys or cross-sectional studies in non-clinical populations or using publicly available social media postings about mental health from unknown users, with inconsistent findings [7,8]. Two recent systematic reviews of internet and social media use highlighted that exposure to self-harm content was associated with normalisation of self-harm behaviour, with potential for harm from triggering, competition, or contagion [9,10]. However, both also cited possible benefits of crisis advice and support on stopping self-harming, reduced social isolation, the potential for therapy and outreach by health professionals. In another review, all 15 studies demonstrated harmful effects from viewing self-harm and suicide-related images, however protective mechanisms were similarly reported in nine of the studies, including self-harm mitigation or reduction and promotion of self-harm recovery [11].

Most existing data on social media or smartphone use and mental health problems in young people, including self-harm, are from cross-sectional studies, some of which have repeated waves [12,13]. Most studies have considered self-reported time spent on social media, rather than the nature or quality of activity, or objective measures of online activity. For smartphone use, dysfunctional behaviours have been evaluated using validated self-report scales of

problematic use (the most common being the Smartphone Addiction Scale [14]), frequency of use or motivations and attitudes [15].

Beyond self-report assessments, there is growing interest in the use of natural language processing (NLP) techniques to extract relevant information from social media data to provide insights into human behaviour. One computational study created a combined dataset of users who had donated their social media data, along with self-report data on past suicide attempts, and users who posted publicly available content describing past suicide attempts on social media, including the date of the suicide attempt [16]. The authors used natural language processing to demonstrate that there are quantifiable signals present in the language used in social media postings that can predict risk for a suicide attempt, with relatively high precision when validated against self-report or social media [16]. Yet the monitoring of language in social media over time is still relatively novel [17,18]. Smartphone tracking has been used to examine specific objective variables, (e.g. daily minutes of screen time and number of phone screen unlocks) over a short observation period (e.g. a week) and the association with mental health symptoms [19]. Yet this has not been widened to study effects on mental health or behaviour over time, such as episodes of self-harm.

What is lacking in self-harm research to date is detailed and frequently collected prospective data about smartphone use experiences, mental health and self-harm, with temporal analyses of objective measures of social media (data upload) and smartphone use (e.g., intensity of usage by time of day and app usage) in those periods before self-harm episodes. To this end, we established the Social media, Smartphone use and Self-harm in Young People (3S-YP) study to prospectively research an enriched clinical cohort of 13–25 year olds using uploaded social media public posts, passive smartphone usage patterns, electronic health records and monthly self-report questionnaires, including outcome data about self-harm, to investigate the mechanistic links.

This paper aims to provide a comprehensive description of the 3S-YP study cohort, including a description of the socio-demographics, self-reported and clinician-recorded history of self-harm, self-reported psychopathology, sleep disturbance, bullying victimisation and loneliness, self-reported social media and smartphone use and electronic health records data at the baseline assessment, as well as an overview of the data available from the baseline and follow up assessments.

## Methods

### Study design and setting

The 3S-YP study used a prospective observational cohort study design. Young people were recruited from South London and Maudsley NHS Foundation Trust (SLaM) between 3rd June 2021 and 30th November 2022 and followed up for 6 months. SLaM provides the widest range of mental health services for children and adults in the UK, including community mental health, home treatment, hospital and outpatient services. SLaM serves a local population of approximately 1.3 million, as well as providing national and specialist services. For full details on the study protocol, see Bye et al. (2023) [20]. We followed the Strengthening the Reporting of Observational studies in Epidemiology (STROBE) Statement to report our cohort study [21] (see S1 File).

### Cohort selection and eligibility

During recruitment, young people were identified from SLaM's Consent for Contact (C4C) patient research participation register. The C4C register comprises of patients who have been approached by a clinician at the Trust to whom they have given their verbal consent to be

contacted about research studies at the Trust for which they may be eligible [22]. Eligibility criteria were applied to the C4C register using the Clinical Research Interactive Search (CRIS) system to identify potentially eligible young people [23]. Researchers confirmed eligibility and extracted contact details from the electronic health records (EHR) using SLaM's electronic Patient Journey System (ePJS). Researchers informed the clinical care team of the intention to approach so they could advise if inappropriate.

To be eligible to participate, young people had to be identified from the C4C register, aged 13–25 years at the time of approach, have accessed Trust services within the last year, and have capacity to consent (and a parent or carer for young people aged 13–15 years). Mental capacity was assumed in the absence of evidence to suggest otherwise, in which case the study protocol was followed. Young people were ineligible if they were unable to complete questionnaires via a smartphone app or online survey platform, admitted to an inpatient psychiatric ward, sectioned under the Mental Health Act (1983) (i.e., legislation to cover the lawful detaining and treatment of an individual without their consent if they present a risk to themselves or other people) or in prison at the time of approach, or a clinician advised it was not appropriate to approach. To reduce the risk of participant selection bias, all eligible young people were invited to participate until the close of recruitment.

## Cohort recruitment and procedure

Eligible young people aged 16–25 years (and parents and carers for eligible young people aged 13–15 years) were sent an initial mobile text message with a unique web link to an online enrolment system. Where a mobile number was not listed on the EHR, researchers sent the invitation via email or called a landline number to get an alternative means of sending the invitation. The enrolment system provided the written participant information, supplemented by a short co-designed animation outlining the study purpose and participation procedure, and digital assent and consent forms.

The initial approach was followed with a telephone call, text message and/or email no less than a week later. Researchers asked young people (and parents and carers for young people aged 13–15 years) for their preferred method for future communications. Young people (and parents and carers for young people aged 13–15 years) who did not respond to three consecutive contact attempts were not contacted again. To maximise recruitment, researchers worked evenings and weekends to fit around young people's commitments. From our previous research and work with youth experts by experience, we have developed detailed standard operating procedures for screening and approaching potential participants, data collection, and managing risk. Procedural guidelines were followed to ensure consistent recruitment practices and minimise repeated or unsolicited contact attempts. Interested young people aged 16–25 years followed online instructions to confirm consent and enrol in the study. Parents and carers of young people aged 13–15 years opted-in during follow-up contact with the researchers to receive a web link to a parent/carer consent form, following which the young person received a web link to confirm their assent and enrol in the study.

Participants who consented to sharing smartphone data were invited to install the 3S-YP app on their device and complete the baseline questionnaire. The 3S-YP app was designed to continuously extract smartphone metadata and administer the questionnaires throughout follow-up. Participants were otherwise provided with the option of completing questionnaires via an online survey platform–Qualtrics (https://www.qualtrics.com). Following baseline completion, participants received automated reminders on the first and seventh day of each month for the next six months inviting them to complete the monthly questionnaires, in accordance with the schedule presented in S1 Table. To reduce participant burden, participants skipped

the first monthly questionnaire in the schedule if they completed the baseline questionnaire within seven days of the following month. Participants could complete the monthly questionnaires at any time during the first seven days of each month only. To maximise data completion at month 6, participants received additional reminders, including telephone calls (in line with the standard operating procedures), and they could complete the final questionnaire at any time during the data collection period. At present, we have included all valid questionnaire data in our summary of the data available. We will consider how to handle questionnaire data for the prospective analyses where there was a significant delay in that data being provided. Consenting participants were invited to provide their social media data following the baseline, month 3 and month 6 questionnaires. Participants were given detailed written and verbal instructions outlining how to download data from social media accounts and upload files to the system. Participants could provide their social media data at any time during the data collection period. EHR data for consenting participants was extracted following the baseline and month-6 questionnaires between 17[th] November 2022 and 18[th] July 2023. After the final questionnaire, a purposive sample of young people (who had consented to further contact) were invited to participate in a brief telephone interview to evaluate the study processes. Informed consent was provided prior to participation in the interview. Study participation was incentivised with shopping vouchers.

## Ethical approval

The 3S-YP study was approved by the National Research Ethics Service, London–Riverside (ethics ref 20/LO/1187; IRAS ref 269104), as well as by the Joint Research and Development Office of the Institute of Psychiatry, Psychology and Neuroscience and SLaM, and the SLaM CRIS Oversight Committee (refs 20–074 and 21–039). The CRIS system was approved as a data resource for secondary analysis by the National Research Ethics Service, South Central–Oxford C (23/SC/0257). This study is registered on ClinicalTrials.gov (ref NCT04601220). All participants aged 16–25 years old provided informed consent via a digital consent form prior to study enrolment. For young people aged 13–15 years old, parents and carers provided informed consent via a digital consent form and the young person provided informed assent via a digital assent form prior to study enrolment. The forms included separate opt-in consents for the study to collect questionnaire, EHR, social media and smartphone data. Participants confirmed which data they were willing to provide by ticking the relevant opt-in box(es). Consent data for all individuals was recorded using the online enrolment system, including unique identifiers and timestamps.

## Measures

Measurements are outlined below, with further detail provided in S2 File. For this publication we summarise information collected at baseline and data availability over the 6-month follow-up period. More detail on current self-harm, social media uploads, smartphone metadata and interview data will be included in future publications.

**Primary outcome.** *Self-reported self-harm*. Self-reported prior and current self-harm were assessed using the Child and Adolescent Self-harm in Europe (CASE) Study criteria [24]. This measure comprises two items for assessing the presence and type of self-harm. Additional items were included at baseline to capture age (in years) when the individual first self-harmed and last self-harmed before baseline.

*Clinician-recorded self-harm*. Clinician-recorded history of self-harm (i.e., occurring prior to the baseline assessment) was identified through manual inspection of risk assessment forms in the EHR of consenting participants using ePJS. Information from free text and structured fields was used to determine the presence of prior self-harm as a dichotomous outcome. Information from free text fields was used to code the type(s) of self-harm.

Using a similar approach to research by Polling et al. [25,26], clinician-recorded current self-harm was identified using the CRIS system by extracting all free text entries that contained any self-harm-related keywords between baseline and month 6 for consenting participants. Extracted data was re-identified and full entries were manually inspected using ePJS. Entries were coded according to the presence of self-harm, presence of self-harm during the participation period, date when self-harm occurred, and type of self-harm. Following coding, all risk assessment forms completed during the participation period were manually inspected to detect any new events not already coded.

Researchers were trained by AB and RD in the data extraction and coding process. Data extraction and coding of the presence of self-harm was conducted by at least one researcher (AB, GW and AVV), with AB validating all self-harm events and coding date and type of self-harm. Any discrepancies or uncertainties were discussed in consensus meetings with RD, and through further discussion with Polling, another clinical academic field expert.

*Classification of self-harm.* Self-harm was classified in accordance with the National Institute for Health and Care Excellence definition: "intentional self-poisoning or injury, irrespective of the apparent purpose of the act" [27] and previous research outlining clinically defined forms of self-harm [25,26]. Clinically defined self-harm comprised of (1) self-poisoning, (2) self-injury, (3) both self-poisoning and self-injury, and (4) other types of self-harm.

We also employed a broader, more inclusive definition to include behaviours not traditionally considered as self-harm where there was a stated intention to self-harm. The decision to include this broader definition was motivated by a desire to capture events described as self-harm that would otherwise have been omitted had we solely employed a clinical definition. Broadly defined self-harm similarly comprised of (1) self-poisoning, (2) self-injury, and (3) both self-poisoning and self-injury, however (4) other types of self-harm also included any events involving alcohol poisoning, descriptions of other prolonged substance misuse (as distinct from a discrete self-poisoning event), and disordered eating behaviours (e.g. fasting, excessive exercise) if there was a stated intention to self-harm. We did not include any of these additional behaviours if there was not a stated intent to self-harm.

**Secondary outcomes.** *Anxiety symptoms.* Symptoms of anxiety were assessed using the Generalized Anxiety Disorder Scale (GAD-7) [28]. Total scores range between 0–21, with scores of 5, 10, and 15 representing the cut-off points for mild, moderate, and severe anxiety, respectively.

*Depression symptoms.* Symptoms of depression were assessed using the Patient Health Questionnaire (PHQ-9) [29]. Total scores range between 0–27, with scores of 5, 10, 15, and 20 representing the cut-off points for mild, moderate, moderately severe, and severe depression, respectively.

*Sleep disturbance symptoms.* Symptoms of sleep disturbance in 13–17 year olds were assessed using the Paediatric Sleep Disturbance Short Form V1.0 4a [30]. Total raw scores range between 4–20 and standardized using a T-score metric, with T-scores of 56, 60 and 66 representing the cut-off points for mild, moderate, and severe sleep disturbance, respectively.

Symptoms of sleep disturbance in adults (aged ≥18) were assessed using the Patient-Reported Outcomes Measurement Information System (PROMIS) Sleep Disturbance Short Form V1.0 4a [31]. Total raw scores range between 4–20 and standardized using a T-score metric, with T-scores of 55, 60 and 70 representing cut-off points for mild, moderate, and severe sleep disturbance, respectively.

*Bullying victimisation.* Bullying victimisation was assessed using the Eight-Item Bullying Checklist derived from the Revised Olweus Bully/Victim Questionnaire (R-OBVQ)[32,33]. Total scores range between 8–40, with higher scores indicative of bullying victimisation. Binary

indicators of regular traditional and cyber bullying victimisation were calculated using a cut-off of three ("2 or 3 times a month") or greater for any of items 1–6 and 7–8, respectively [33,34].

*Loneliness*. Feelings of loneliness were assessed using the Three-Item Loneliness Scale [35]. Total scores range between 3–9, with higher scores indicative of greater feelings of loneliness.

**Exposures.** *Self-report social media and smartphone use*. Self-reported social media and smartphone use were assessed using unvalidated items. For example, "How much time do you usually spend on social media on weekends?" and "How much time do you usually spend on your phone on weekends?", with response options ranging between "Less than 30 minutes" and "More than 6 hours".

*Problematic smartphone use*. Problematic smartphone use was assessed using the Smartphone Addiction Scale-Short Version (SAS-SV) [36]. Total scores range between 10–60, with a score of ≥31 demonstrating a sensitivity of 0.867 and specificity of 0.893 for smartphone addiction in adolescent males and a score of ≥33 demonstrating a sensitivity of 0.875 and a specificity of 0.886 for smartphone addiction in adolescent females. There is no available published guidance on validated cut-offs for young people who do not identify as either male or female. In this study, for participants who prefer to self-describe their gender, we applied the higher threshold of ≥33 to categorise excessive phone use.

*Social media uploads*. Social media meta-, imagery and textual data were obtained from social media uploads from consenting participants. Individuals' most recent valid data uploads from each platform were processed.

*Smartphone metadata*. Smartphone metadata was extracted continuously using the Google Android App Usage Application Programming Interface (API) for consenting participants. This did not include iPhone users due to data protection restrictions with iOS–a mobile operating system developed by Apple Inc. -.

**Other measurements.** Self-reported sociodemographic and exposure to Covid-19 data were collected. Sociodemographic and clinical data were extracted from consenting participants EHR using the CRIS system. The post-study participation interview topic guide was co-designed with young people to facilitate informal discussions on the experience of participation and perspectives on the research topic more broadly.

## Patient and public involvement (PPI)

We have employed a participatory research approach to promote engagement, inclusiveness and representation in this study. This has been facilitated by co-author/co-investigator and senior service user consultant–SB-F, and our charity partner, leading national UK youth mental health charity–YoungMinds,. SB-F has been a core member of the project group and instrumental in shaping the study from the outset, including contributing to research priority setting, the ethical approval process, study procedures and participant-facing materials including testing the 3S-YP app, and attending Project Steering Group meetings. YoungMinds have facilitated wider engagements through their national youth advisory programme, which has provided further opportunities to work with young people through consultations, focus groups, workshops and the Project Steering Group to co-create the study procedures, participant-facing materials including the 3S-YP app, and analysis and dissemination plans. All participants were advised findings will be available via the study website (www.3syp.com).

## Results

### Recruitment

Fig 1 presents the total number of young people screened for eligibility and reasons for non-participation. The 3S-YP study was open to recruitment for 18 months between 3rd June 2021

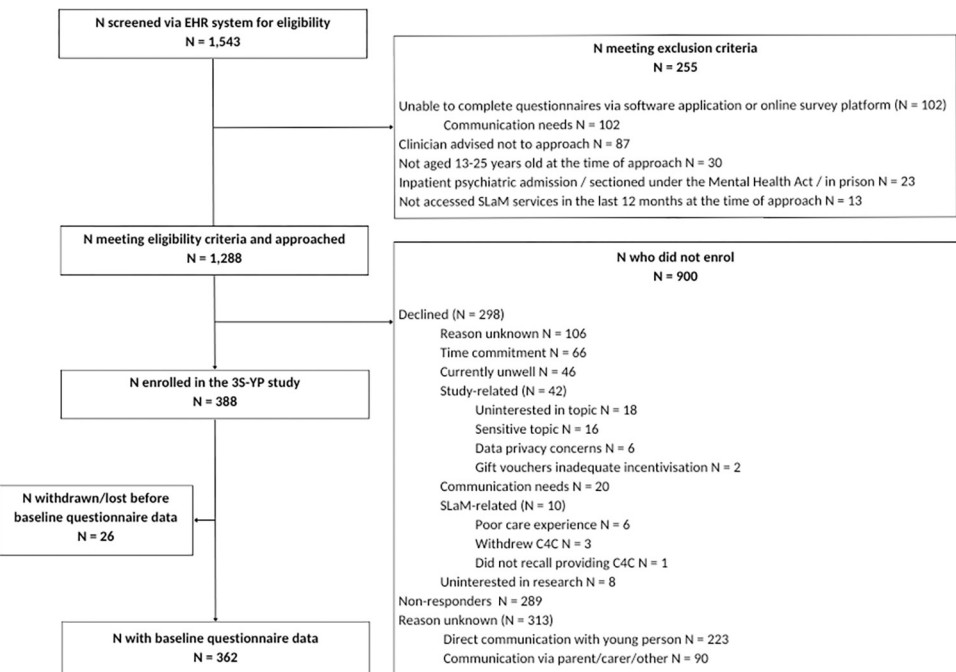

**Fig 1. Recruitment flowchart of young people through the 3S-YP study from screening to baseline completion.**

and 30th November 2022, with the first young person enrolling in the study on the 14th June 2021. During the recruitment period, 1,543 young people were screened for eligibility, of whom 255 did not meet eligibility criteria. The approached population therefore comprised 1,288 eligible young people who were approached to participate. Of the approached population, 388 (30.1%) young people enrolled in the study. Of those enrolled, 362 (93.3%) provided baseline questionnaire data and were followed up for 6 months. The 3S-YP cohort was similar to the approached population regarding age, ethnicity and primary diagnosis but had a larger proportion of female participants (70.2% female in the cohort compared to 61.3% in the approached population) (see S2 Table).

## Sociodemographic characteristics

Table 1 presents the self-reported sociodemographic information for the 3S-YP cohort. The majority were aged ≥18 years (n = 244; 67.4%) and female (n = 225; 62.2%). Nearly 10% (n = 29; 8.0%) of young people preferred to self-describe their gender, including those who identified as non-binary, trans and gender fluid. The cohort was ethnically diverse, with over 30% of the sample self-identifying as being from Black (n = 51; 14.1%) or Mixed/Multiple ethnic backgrounds (n = 64; 17.7%). Most young people were in some form of education (n = 232; 64.1%) and had achieved GCSE grades 4 or above (n = 268; 74.0%). One third were in employment (n = 108; 29.8%).

Approximately half the cohort had ever smoked (n = 195; 53.9%), used e-cigarettes (n = 196, 54.1%) and tried drugs (n = 185; 51.1%). Around a third were current smokers (n = 110; 30.4%) and e-cigarette users (n = 105; 29.0%). Most young people had ever had an alcoholic drink (n = 289; 79.8%), with half usually drinking alcohol at least once a month (n = 189; 52.2%). Young people were a mean age of 14.8 (SD 2.9) and 14.3 (SD 4.3) years when they first tried smoking and had an alcoholic drink, respectively.

**Table 1. Self-reported sociodemographic characteristics at the baseline assessment.**

| Variable | | Cohort (N = 362) n (%) |
|---|---|---|
| **Age in years** | | Mean (SD): 19.2 (3.2) |
| **Age categories** | | |
| | 13–15 | 39 (10.8) |
| | 16–17 | 79 (21.8) |
| | 18–21 | 155 (42.8) |
| | 22–25+ | 89 (24.6) |
| | Missing | - |
| **Gender** | | |
| | Female | 225 (62.2) |
| | Male | 106 (29.3) |
| | Prefer to self-describe | 29 (8.0) |
| | Missing | 2 (0.6) |
| **Ethnicity** | | |
| | Any Asian or Asian British background | 23 (6.4) |
| | Any Black, African, Caribbean or Black British background | 51 (14.1) |
| | Any Mixed or Multiple ethnic background | 64 (17.7) |
| | Any White background | 212 (58.6) |
| | Any other ethnic group | 10 (2.8) |
| | Missing | 2 (0.6) |
| **Educationstatus** | | |
| | Secondary school | 38 (10.5) |
| | Sixth form/college | 103 (28.5) |
| | University | 74 (20.4) |
| | Other (e.g. home tuition/pupil referral unit) | 17 (4.7) |
| | None of the above | 126 (34.8) |
| | Missing | 4 (1.1) |
| **Employment status[a]** | | |
| | Employed/self-employed | 108 (29.8) |
| | Student | 232 (64.1) |
| | Not working for health reasons | 51 (14.1) |
| | Unemployed | 33 (9.1) |
| | Other(e.g. carer/homemaker) | 19 (5.2) |
| | Missing | 10 (2.8) |
| **Highest level of education** | | |
| | No formal qualifications | 41 (11.3) |
| | GCSE grades 3-1/D-G, NVQ level 1 or equivalent | 23 (6.4) |
| | GCSE grades 9-4/A*-C, NVQ level 2 or equivalent | 111 (30.7) |
| | AS/A Level, NVQ level 3 or equivalent | 112 (30.9) |
| | Diploma level or above | 45 (12.5) |
| | Other | 18 (5.0) |
| | Missing | 12 (3.3) |
| **Smoking** | | |
| | Never smoked | 163 (45.0) |
| | Tried smoking | 43 (11.9) |
| | Past smoker | 42 (11.6) |
| | Occasional smoker | 41 (11.3) |

*(Continued)*

**Table 1.** (Continued)

| Variable | | Cohort (N = 362) n (%) |
|---|---|---|
| | Regular smoker | 69 (19.1) |
| | Missing | 4 (1.1) |
| **E-cigarette use** | | |
| | Never used e-cigarettes | 160 (44.2) |
| | Tried using e-cigarettes | 53 (14.6) |
| | Past e-cigarette user | 38 (10.5) |
| | Occasional e-cigarette user | 35 (9.7) |
| | Regular e-cigarette user | 70 (19.3) |
| | Missing | 6 (1.7) |
| **Ever had an alcoholic drink** | | |
| | Yes | 289 (79.8) |
| | No | 72 (19.9) |
| | Missing | 1 (0.3) |
| **Usual frequency of drinking alcohol** | | |
| | At least once a week | 115 (31.8) |
| | About once a fortnight to once a month | 74 (20.4) |
| | Only a few times a year | 65 (18.0) |
| | Do not drink alcohol | 106 (29.3) |
| | Missing | 2 (0.6) |
| **Ever taken drugs** | | |
| | Yes | 185 (51.1) |
| | No | 168 (46.4) |
| | Missing | 9 (2.5) |
| **Exposure to Covid-19** | | |
| | Yes (i.e., positive test or suspected exposure) | 187 (51.7) |
| | Unsure/declined | 45 (12.4) |
| | No | 125 (34.5) |
| | Missing | 5 (1.4) |

[a]Young people were able to select all response options that were relevant.

## Self-reported and clinician-recorded history of self-harm

Above 80% of the cohort had self-reported or clinician-recorded history of self-harm, according to clinical (n = 295, 81.5%) and broad definitions of self-harm (n = 296, 81.8%) (see Table 2). Rates of prior self-harm from self-report and clinician-recorded data were not dissimilar, with clinician-recorded rates slightly higher. Young people reported they were a mean age of 12.5 (SD 3.1) when they first self-harmed and 18.5 (SD 2.9) years when they last self-harmed.

## Self-reported psychopathology, sleep disturbance, bullying victimisation and loneliness

The majority of young people had baseline scores for moderate or severe anxiety (n = 244; 67.4%) and depression (n = 255; 70.4) (see Table 3). Approximately 40% reported met the cut-off for moderate or severe sleep disturbance (n = 171; 47.2%). To note, two young people had their 18th birthday between being invited to participate and enrolment and thus were invited

**Table 2. Self-reported and clinician-recorded history of self-harm at the baseline assessment.**

| Variable | | Cohort (N = 362) n (%) |
|---|---|---|
| **Self-reported self-harm** | | |
| **History of self-harm** | | |
| | No | 109 (30.1) |
| | Yes, once | 33 (9.1) |
| | Yes, more than once | 216 (59.7) |
| | Missing | 4 (1.1) |
| **Clinically defined history of self-harm** | | |
| | No | 109 (30.1) |
| | Yes | 249 (68.8) |
| | Missing | 4 (1.1) |
| **Broadly defined history of self-harm** | | |
| | No | 109 (30.1) |
| | Yes | 249 (68.8) |
| | Missing | 4 (1.1) |
| **Clinician-recorded self-harm** | | |
| **Clinically defined history of self-harm** | | |
| | No | 78 (21.5) |
| | Yes | 264 (72.9) |
| | Missing | 20 (5.5) |
| **Broadly defined history of self-harm** | | |
| | No | 77 (21.3) |
| | Yes | 265 (73.2) |
| | Missing | 20 (5.5) |
| **Self-reported/clinician-recorded self-harm** | | |
| **Clinically defined history of self-harm** | | |
| | No | 59 (16.3) |
| | Yes | 295 (81.5) |
| | Missing | 8 (2.2) |
| **Broadly defined history of self-harm** | | |
| | No | 58 (16.0) |
| | Yes | 296 (81.8) |
| | Missing | 8 (2.2) |

to complete the Pediatric Sleep Disturbance Short Form at baseline and for the duration of the follow up period. About a third of young people had experienced regular bullying victimisation (n = 129; 35.6%), but less than 5% had experienced regular cyber bullying (n = 16; 4.4%). The mean score on the Three-Item Loneliness Scale (35) were 6.8 (SD 2.0) from a possible 9.

## Self-reported social media and smartphone use

Table 4 presents self-reported social media and smartphone use at the baseline assessment. Nearly all young people reported using social media at baseline (n = 347; 95.9%), with Instagram (n = 310; 85.6%) and YouTube (n = 297; 82.0%) the most frequently used. Social media was used for a variety of purposes, most commonly direct messaging (n = 308; 85.1%) or liking/commenting on a post (n = 285; 78.7%). Around a quarter usually spent more than 5 hours on social media on weekdays (n = 88; 24.3%), slightly less than on weekends (n = 116;

**Table 3. Self-reported psychopathology, sleep disturbance, bullying victimisation, and loneliness at the baseline assessment.**

| Variable | Cohort (N = 362) n (%) |
|---|---|
| **Generalised Anxiety Disorder (GAD-7)** | Mean (SD): 12.8 (6.1) |
| Minimal anxiety | 42 (11.6) |
| Mild anxiety | 74 (20.4) |
| Moderate anxiety | 84 (23.2) |
| Severe anxiety | 160 (44.2) |
| Missing | 2 (0.6) |
| **Patient Heath Questionnaire (PHQ-9)** | Mean (SD): 14.7 (7.6) |
| Minimal depression | 38 (10.5) |
| Mild depression | 67 (18.5) |
| Moderate depression | 66 (18.2) |
| Moderately severe depression | 67 (18.5) |
| Severe depression | 122 (33.7) |
| Missing | 2 (0.6) |
| **PROMIS Pediatric Sleep Disturbance Short Form (n = 120)** | Mean (SD): 61.1 (12.0) |
| Within normal limits | 38 (31.7) |
| Mild sleep disturbance | 5 (4.2) |
| Moderate sleep disturbance | 29 (24.2) |
| Severe sleep disturbance | 47 (39.2) |
| Missing | 1 (0.8) |
| **PROMIS Sleep Disturbance Short Form (n = 242)** | Mean (SD): 57.4 (11.7) |
| Within normal limits | 96 (39.7) |
| Mild sleep disturbance | 48 (19.8) |
| Moderate sleep disturbance | 66 (27.3) |
| Severe sleep disturbance | 29 (12.0) |
| Missing | 3 (1.2) |
| **Eight-Item Bullying Checklist** | |
| Total score | Mean (SD): 11.5 (4.8) |
| Missing | 13 (3.6) |
| Traditional bullying | |
| No | 229 (63.3) |
| Yes | 129 (35.6) |
| Missing | 4 (1.1) |
| Cyberbullying | |
| No | 336 (92.8) |
| Yes | 16 (4.4) |
| Missing | 10 (2.8) |
| **Three-Item Loneliness scale** | Mean (SD): 6.8 (2.0) |
| Missing | 8 (2.2) |

32.0%). More than 50% of young people reported using social media after midnight on weekdays (n = 197; 54.4%) and this rose by 10% on weekends (n = 241; 66.6%).

Young people used a range of apps on their phones, most commonly social media apps (n = 186; 51.4%). About 40% reported spending more than 5 hours on their phone on weekdays (n = 152; 42.0%) and 50% on weekends (n = 182; 50.3%). A substantial proportion used their phone after midnight on weekdays (n = 215; 59.4%) and weekends (n = 263; 72.7%).

**Table 4. Self-reported social media and smartphone use at the baseline assessment.**

| Variable | | Cohort (N = 362) n (%) |
|---|---|---|
| **Social media use** | | |
| **Use of social media** | | |
| | Yes | 347 (95.9) |
| | No | 13 (3.6) |
| | Missing | 2 (0.6) |
| **Most frequently used platform[a]** | | |
| | Facebook | 180 (49.7) |
| | Instagram | 310 (85.6) |
| | Pinterest | 135 (37.3) |
| | Snapchat | 248 (68.5) |
| | TikTok | 248 (68.5) |
| | Twitter | 131 (36.2) |
| | YouTube | 297 (82.0) |
| | Other | 81 (22.4) |
| | Missing | 16 (4.4) |
| **Main purpose of usage[a]** | | |
| | Direct message | 308 (85.1) |
| | Like/comment on a post | 285 (78.7) |
| | Post on own page/timeline/story/blog | 236 (65.2) |
| | Share on own page/timeline/story/blog | 223 (61.6) |
| | View page/timeline/story/blog of a friend | 277 (76.5) |
| | View page/timeline/story/blog of someone unknown (e.g., celebrity) | 244 (67.4) |
| | Scroll through news feed | 260 (71.8) |
| | Other | 34 (9.4) |
| | Missing | 15 (4.1) |
| **Average daily usage on weekdays** | | |
| | Non-user | 13 (3.6) |
| | Less than 1 hour | 34 (9.4) |
| | Between 1 to 3 hours | 129 (35.6) |
| | Between 3 to 5 hours | 94 (26.0) |
| | >5 hours | 88 (24.3) |
| | Missing | 4 (1.1) |
| **Average daily usage on weekends** | | |
| | Non-user | 13 (3.6) |
| | Less than 1 hour | 20 (5.5) |
| | Between 1 to 3 hours | 102 (28.2) |
| | Between 3 to 5 hours | 97 (26.8) |
| | >5 hours | 116 (32.0) |
| | Missing | 14 (3.9) |
| **Latest time of weekday use** | | |
| | Non-user | 13 (3.6) |
| | Before 10pm | 52 (14.4) |
| | Between 10pm and midnight | 94 (26.0) |
| | Between midnight and 2am | 112 (30.9) |
| | After 2am | 85 (23.5) |
| | Missing | 6 (1.7) |

(*Continued*)

**Table 4.** (Continued)

| Variable | Cohort (N = 362)<br>n (%) |
|---|---:|
| **Latest time of weekend use** | |
| Non-user | 13 (3.6) |
| Before 10pm | 33 (9.1) |
| Between 10pm and midnight | 65 (18.0) |
| Between midnight and 2am | 100 (27.6) |
| After 2am | 141 (39.0) |
| Missing | 10 (2.8) |
| **Smartphone use** | |
| **Most frequently used app** | |
| Social media | 186 (51.4) |
| Games | 23 (6.4) |
| Messaging | 57 (15.7) |
| Entertainment | 63 (17.4) |
| Email | 10 (2.8) |
| Other | 18 (5.0) |
| Missing | 5 (1.4) |
| **Average daily usage on weekdays** | |
| Non-user | - |
| Less than 1 hour | 14 (3.9) |
| Between 1 to 3 hours | 73 (20.2) |
| Between 3 to 5 hours | 112 (30.9) |
| >5 hours | 152 (42.0) |
| Missing | 11 (3.0) |
| **Average daily usage on weekends** | |
| Non-user | - |
| Less than 1 hour | 8 (2.2) |
| Between 1 to 3 hours | 64 (17.7) |
| Between 3 to 5 hours | 95 (26.2) |
| >5 hours | 182 (50.3) |
| Missing | 13 (3.6) |
| **Latest time of weekday use** | |
| Non-user | - |
| Before 10pm | 43 (11.9) |
| Between 10pm and midnight | 96 (26.5) |
| Between midnight and 2am | 119 (32.9) |
| After 2am | 96 (26.5) |
| Missing | 8 (2.2) |
| **Latest time of weekend use** | |
| Non-user | - |
| Before 10pm | 28 (7.7) |
| Between 10pm and midnight | 60 (16.6) |
| Between midnight and 2am | 115 (31.8) |
| After 2am | 148 (40.9) |
| Missing | 11 (3.0) |
| **Use phone at mealtimes** | |
| Yes | 197 (54.4) |

(*Continued*)

**Table 4.** (Continued)

| Variable | | | Cohort (N = 362)<br>n (%) |
|---|---|---|---|
| | No | | 159 (43.9) |
| | Missing | | 6 (1.7) |
| **Phone in bedroom at night** | | | |
| | Yes | | 336 (92.8) |
| | No | | 19 (5.2) |
| | Missing | | 7 (1.9) |
| **Nighttime power mode** | | | |
| | On | | 94 (26.0) |
| | Silent | | 229 (63.3) |
| | Off | | 15 (4.1) |
| | Missing | | 24 (6.6) |
| **Self-evaluation of excessive use** | | | |
| | Yes | | 266 (73.5) |
| | No | | 49 (13.5) |
| | Unsure | | 41 (11.3) |
| | Missing | | 6 (1.7) |
| **Smartphone Addiction Scale—Short Version (SAS-SV)** | | | Mean (SD): 32.6 (9.9) |
| | Not problematic smartphone use | | 179 (49.4) |
| | Problematic smartphone use | | 177 (48.9) |
| | Missing | | 6 (1.7) |

[a]Young people were able to select all response options that were relevant.

Young people often used their phone at mealtimes (n = 197; 54.4%) and had access to it during the night (n = 336; 92.8%), with a quarter leaving it on but not on silent mode (n = 94; 26.0%). Approximately 70% self-evaluated their phone use as excessive (n = 266; 73.5%) and almost half met the cut-off for problematic smartphone use on the SAS-SV (36) (n = 177; 48.9%).

## EHR data at the baseline assessment

Table 5 presents the EHR data at the baseline assessment for participants who consented to EHR data access (n = 345). Of those consenting, median IMD rank was 13168 (IQR 7294–

**Table 5.  Electronic health records data at the baseline assessment.**

| Variable | | | Cohort (N = 345)<br>n (%) |
|---|---|---|---|
| **Local quartiles of multiple deprivation** | | | |
| | | 1 (most deprived) | 104 (30.1) |
| | | 2 | 112 (32.5) |
| | | 3 | 75 (21.7) |
| | | 4 (least deprived) | 42 (12.2) |
| | | Unknown | 12 (3.5) |
| **Number of years since first accepted SLaM referral** | | | Mean (SD): 5.2 (3.8)<br>Median (IQR): 4.5 (2.0–8.0) |
| **Prior psychiatric inpatient admission** | | | 43 (12.5 |
| **Previous section under the Mental Health Act** | | | 21 (6.1) |

19109) and approximately a third lived in the most deprived areas in England at the baseline assessment (n = 104; 30.1%). Young people had their first referral to SLaM accepted a mean of 5.2 (SD 3.8) years prior to baseline. More than 10% had previously been admitted to a psychiatric inpatient ward (n = 43; 12.5%) and 6% had been subject to a section under the Mental Health Act (n = 21; 6.1%).

### Data availability

Fig 2 illustrates the flow of participants through the 3S-YP study. The number of participants providing any questionnaire data at each time point (relative to the number of active participants, i.e., after excluding those who withdrew or discontinued providing questionnaire data) was month 1: n = 189/240 (78.8%); month 2: n = 219/294 (74.5%); month 3: n = 185/273 (67.8%), month 4: n = 162/262 (61.8%), month 5: n = 146/248 (58.9%); month 6: n = 230/230 (100.0%) (see S3 Table for further detail). Notably, the number invited for month 1 also excludes 71 active participants who completed their baseline questionnaire within seven days of the next month, so did not receive the month 1 questionnaire.

Most of the cohort (n = 345/362; 95.3%) consented to EHR data access and had sociodemographic and clinical data available in the EHR. Most participants (n = 283/362; 78.2%) agreed to share their social media data, and of those, we have processed data available for 110 (38.9%). Similarly, most participants (n = 312/362; 86.2%) consented to sharing smartphone data, and of those, 287 (92.0%) installed the smartphone app at baseline (iOS n = 218, 76.0%; Android n = 69, 24.0%). Of the 69 who installed the Android version of the app, metadata was available (at any time during participation) from 48 (69.6%) young people. We also have interview data available for 16 young people who participated in the 3S-YP study.

### Discussion

The 3S-YP study is a novel prospective study with a 6-month follow up of adolescents and young adults who have had mental health issues requiring secondary mental healthcare. It was designed to investigate patterns of social media and smartphone use in association with self-harm and other relevant outcomes. Prior research is largely limited to cross-sectional survey data, demonstrating associations [37,38] rather than offering insight into potential underlying mechanisms, and social media analyses in computational epidemiology studies that select data for a specific outcome. For example, studying depression in users who post about depression [39], instead of first selecting a population and then measuring outcomes [40]. In this study we have used a wide range of standardised self-report measures and have the unique perspective of linkage with EHRs and user-generated data, as well as qualitative interview data. With the breadth of data generated, we hope to be able to extend current understanding from associations in cross-sectional survey data [37,38] to a more nuanced understanding of the impact on young people and the role of mediating factors. This will be vital for informing the development of interventions. In this paper, we have provided a comprehensive description of the cohort baseline data and an overview of the data availability at baseline and during follow up.

In our cohort, we observed high rates of historical self-harm in both self- and clinician-recorded data. The rates are considerably higher compared to community studies—which have reported aggregate lifetime risk of self-harm in young people between 11.0–17.0% globally [41] - although there is substantial variation in the literature owing to methodological differences between studies [41–44]. In clinical populations with known mental health problems, it is unsurprising that these risks are higher [45]. We also found that rates of reporting were marginally higher in clinician-reported data than self-reported data. This discrepancy is in line with one of the few other studies that have combined these two sources of data, though in a

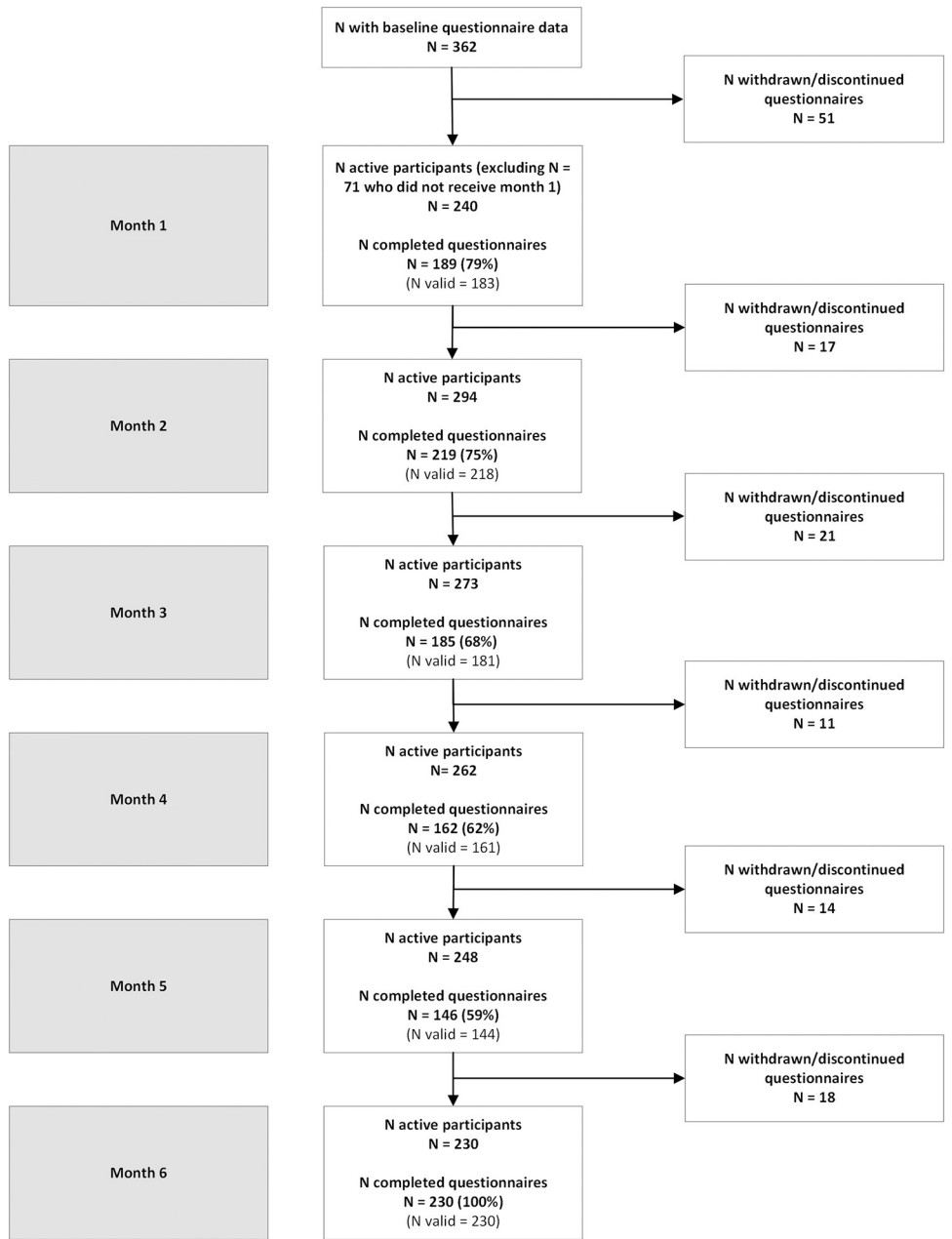

**Fig 2. Participation flowchart of young people through the 3S-YP study from baseline completion to month 6.**

non-clinical population [42]. Studies on youth self-harm typically use either self-report or EHR data, and it is widely accepted that both sources have their limitations. Self-report data can be subject to recall and response bias [42,46]. Whilst EHR data is limited to self-harm that comes to the attention of health services and is dependent on data completeness and the quality of data reporting relevant to the research question [47,48]. By using both sources of data and the thorough approach to EHR data extraction specifically, we have sought to capture the broad spectrum of self-harm behaviours occurring in this population. The findings highlight the need for increased investment in prevention and early intervention work to reduce the risk of self-harm occurring in young people at risk of developing mental health issues.

Furthermore, screening youth service users for self-harm and providing treatment in accordance with clinical guidance [27] to reduce the risk of repeat self-harm or an escalation of allied behaviours is paramount, especially as we know that self-harm is the most significant risk factor for suicide [1,2].

Symptoms of moderate to severe current anxiety, depression and sleep disturbance at the baseline assessment were common. These findings are higher than reports from non-clinical populations [49,50] and to be expected given these symptoms are more commonly experienced by individuals with mental health difficulties [51]. About a third of young people had recently been the victim of bullying, which may further exacerbate any mental health difficulties. Traditional methods of bullying (e.g., social exclusion) were more frequently reported than cyberbullying and cyberbullying was lower than expected considering prior research and the profile of our cohort [52,53], in particular the high levels of digital technology usage. Mean baseline scores on the Three-Item Loneliness Scale were higher than reports from general population data [54], but correspond to other clinical population studies, suggesting it may relate to impairment in social functioning that can be symptomatic of some mental health conditions [55]. Loneliness may also be exacerbated by excessive time spent online, that would be otherwise spent investing in offline social networks [56,57].

In the present cohort, one in two young people reportedly spent excessive time (i.e. > 5 hours) on their smartphone at weekends and were classified with problematic smartphone use. We are not aware of another study that has investigated problematic smartphone use in clinical samples. Findings from a systematic review and meta-analysis of non-clinical studies reported lower prevalence of problematic smartphone use of between 10 and 30% of children and young people [58]. But given the evidenced associations with depression, anxiety and sleep disturbance [58], all of which were prevalent among our cohort at the baseline assessment, our rates are unsurprisingly higher. Interestingly, recent research, conducted in the same time frame as the present study, investigated the impact of the COVID-19 pandemic on student mental health. It showed that rates of problematic smartphone use may be increasing as a result [59]. Nonetheless, it is difficult to know what scales are measuring since devices have a range of purposes and in turn, what aspects need targeting in interventions, though there is some evidence, including in our cohort, that youth smartphone usage may be largely driven by social media use [60]. Nearly all young people reported using some form of social media, typically for direct messaging, liking, commenting on and viewing content. A quarter reported spending excessive time (i.e. > 5 hours) on platforms on weekdays and more so at weekends. As with smartphone use, we lack comparable clinical research data, but the high usage reported in our cohort is greater than findings from a large nationally representative UK birth cohort study [38]. Scott et al. (2019) [38] also suggested a possible link with sleep disturbance, which was similarly more prevalent among our cohort. The findings reported to date lend themselves to discussions around the development and evaluation of interventions to address problematic digital technology use in clinical youth populations, of which there is some evidence of effectiveness [61].

## Strengths and limitations

This is the first prospective study to investigate the impact of digital technology use on self-harm in a clinical youth cohort. This study has generated one of the most comprehensive databases based on our clinical cohort data, offering a wealth of opportunities to explore the data from a range of methodological approaches, with data available from self-report measures, EHRs and qualitative interviews as well as more objective data sources, i.e., user-generated data. As a multidisciplinary team, we have incorporated a range of perspectives in the research

planning, including lived-experience, clinical academic psychiatry, and academic psychology, as well as computational, statistical, and qualitative approaches. We have also had extensive youth involvement, facilitated by co-author (SB-F) and the leading national UK youth mental health charity–YoungMinds, in the design and conduct of this research and will continue to do so, to enhance the acceptability, appropriateness, and relevance [62]. Furthermore, remote recruitment via an NHS-led research participation register and wide inclusion criteria provided an efficient means of screening potentially eligible young people [63] and yielded a heterogeneous sample of adolescents and young adults who were broadly representative of the approached population.

There are also several potential limitations to the 3S-YP study. Missing data is common in prospective studies, particularly among clinical samples. The flexible approach to participation in this study provided young people with greater autonomy in the decision to participate but may have further increased the risk of missing and inconsistent data across the different data sources over time. It is important to note that this study is novel and exploratory, accessing user-generated smartphone and social media data that is largely restricted for use by academic researchers, even where consent has been obtained, resulting in considerable participant burden for young people willing to share their data [64]. Although using apps and web-based systems for data collection, rather than traditional methods, e.g. in-person, has many advantages for researchers and participants, it may have contributed to the limited response rates over time [65]. A proportion of the sample were invited to complete fewer assessments in total compared to the rest of the sample, depending on the date they completed the baseline questionnaire. The effects of which will be considered in future analyses of the prospective data to explore possible implications for data availability and retention. Although a single recruitment site can limit generalisability, SLaM provides national and local services, so the sample is not limited to a geographical catchment area. There may be systematic differences between the C4C population, of whom we approached for this study, and the wider SLaM patient population as individual and service-related factors may influence who is approached and who consents to join the C4C register [66]. The varying clinical severity among our cohort and the need to communicate in English and be digitally literate to participate may also have increased the risk of selection bias.

## Conclusion

The 3S-YP study is the first prospective study with a 6-month follow-up of a clinical youth sample, in which patterns of social media and smartphone use in relation to self-harm behaviour were investigated. It is also the first study to use novel data linkages, i.e., self-report, EHR and user-generated data, to produce a comprehensive database from which to investigate the impact of digital technology on youth mental health. Findings from the baseline assessment indicate self-harm, anxiety, depression, sleep disturbance and digital technology overuse are common among adolescent and young adult service users. These findings emphasise the importance of increased investment in prevention and early intervention to reduce the risk of self-harm occurring in at-risk youth. Findings also add weight to screening patients for self-harm and interventions for young people with a history of self-harm to reduce the risk of repeat or an escalation in self-harm, as well as digital technology use interventions for clinical youth populations for whom social media use has become problematic. Future analyses will be important for understanding the mechanisms underpinning associations between digital technology use and youth mental health over time, as well as comparisons between self-report and user-generated data.

## Supporting information

**S1 Table. Outline of data collection schedule for the 6-month follow-up period.**
(DOCX)

**S2 Table. Characteristics of approached population and enrolments from EHR data at screening.**
(DOCX)

**S3 Table. Data availabilityfor self-reported measures from baseline and follow up assessments for the total cohort.**
(DOCX)

**S1 File. STROBE statement—checklist of items that should be included in reports of observational studies.**
(DOCX)

**S2 File. Detailed description of measures.**
(DOCX)

## Acknowledgments

We would like to thank the young people who participated in this study, and their parents, carers, and other family members for their support. We would like to thank the steering group members, including Louise Arsenault, Janis Baird, Johnny Downs, Tamsin Ford, Nuala Flewett, Paul Gringras, Matthew Hotopf, Hazel Inskip, Nicola Kalk, Navneet Kapur, Fiona Lacey, Dennis Ougrin, Louise Pratt, Angus Roberts, Hannah Russell, Edmund Sonuga-Barke, Robert Stewart, Jack Stonebridge, Alastair Sutcliffe, Sumithra Velupillai and Tony Wood, domain expert Catherine Polling, and the young people who contributed to the study development and progress. We would like to thank YoungMinds for partnering with us on this work. We would like to thank Jack Murray, SLaM research nurse, for support with recruitment. We would also like to thank the CRIS team for their support with recruitment and data extraction, including Matthew Broadbent, Pampa Chakravarti, Debbie Cummings, Amelia Jewell, Daisy Kornblum and Megan Pritchard.

## Author Contributions

**Conceptualization:** Ben Carter, Kylee Trevillion, Maria Liakata, Stella Branthonne-Foster, Rina Dutta.

**Data curation:** Amanda Bye, Daniel Leightley, Samantha Cross, Zohra Zenasni, Rina Dutta.

**Formal analysis:** Amanda Bye, Samantha Cross, Zohra Zenasni, Ewan Carr, Rina Dutta.

**Funding acquisition:** Ben Carter, Kylee Trevillion, Maria Liakata, Stella Branthonne-Foster, Rina Dutta.

**Investigation:** Amanda Bye, Zohra Zenasni, Grace Williamson, Alba Vega Viyuela.

**Methodology:** Amanda Bye, Ben Carter, Daniel Leightley, Kylee Trevillion, Maria Liakata, Stella Branthonne-Foster, Rina Dutta.

**Project administration:** Amanda Bye.

**Resources:** Amanda Bye, Daniel Leightley.

**Software:** Amanda Bye, Daniel Leightley, Stella Branthonne-Foster, Grace Williamson, Rina Dutta.

**Supervision:** Amanda Bye, Ben Carter, Rina Dutta.

**Validation:** Amanda Bye, Ben Carter, Rina Dutta.

**Visualization:** Amanda Bye, Ben Carter, Samantha Cross, Rina Dutta.

**Writing – original draft:** Amanda Bye, Rina Dutta.

**Writing – review & editing:** Amanda Bye, Ben Carter, Daniel Leightley, Kylee Trevillion, Maria Liakata, Stella Branthonne-Foster, Samantha Cross, Zohra Zenasni, Ewan Carr, Grace Williamson, Alba Vega Viyuela, Rina Dutta.

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
