## [Decision Letter · Decision Letter 0]

2 Jan 2024

PONE-D-23-38299Cohort profile: The Social media, Smartphone use and Self-harm in Young People (3S-YP) study – a prospective, observational cohort study of young people in contact with mental health servicesPLOS ONE

Dear Dr. Bye,

Thank you for submitting your manuscript to PLOS ONE. After careful consideration, we feel that it has merit but does not fully meet PLOS ONE’s publication criteria as it currently stands. Therefore, we invite you to submit a revised version of the manuscript that addresses the points raised during the review process.

We look forward to receiving your revised manuscript.

Kind regards,

Vincenzo De Luca

Academic Editor

PLOS ONE

 [This work was supported by the Medical Research Council and Medical Research Foundation (grant number MR/S020365/1). This work was also part supported by the National Institute for Health and Care Research (NIHR) Maudsley Biomedical Research Centre (BRC) and King’s College London, and the NIHR Clinical Research Network (CRN) South London. RD was also funded by a Clinician Scientist Fellowship from the Health Foundation in partnership with the Academy of Medical Sciences and her work is supported by the National Institute for Health Research (NIHR) Biomedical Research Centre at South London and Maudsley NHS Foundation Trust and King’s College London. BC is also supported by the Nuffield Trust. ML is also supported by the Engineering and Physical Sciences Research Council (grant number EP/V030302/1) and The Alan Turing Institute (grant number EP/N510129/1). AB and EC are also supported by the National Institute for Health and Care Research (NIHR) Maudsley Biomedical Research Centre (BRC) and King’s College London. The views expressed are those of the author(s) and not necessarily those of the MRC, the MRF, the NHS, the NIHR or the Department of Health and Social Care.  

For the purposes of open access, the author has applied a Creative Commons Attribution (CC BY) licence to any Accepted Author Manuscript version arising from this submission].  

5. In the online submission form, you indicated that [All data are stored and managed by the 3S-YP research team at King’s College London. Data supporting this study cannot be placed in a public depository due to ethical and data protection restrictions. De-identified data from this study can be made available upon reasonable request to interested collaborators. For any further information or potential collaboration, requests can be directed to the Project Manager, Dr Amanda Bye (amanda.bye@kcl.ac.uk) and the Chief Investigator, Dr Rina Dutta (rina.dutta@kcl.ac.uk).]. 

Reviewers' comments:

Reviewer's Responses to Questions

**Comments to the Author**

1. Is the manuscript technically sound, and do the data support the conclusions?

Reviewer #1: Yes

Reviewer #2: Yes

2. Has the statistical analysis been performed appropriately and rigorously? 

Reviewer #1: Yes

Reviewer #2: N/A

3. Have the authors made all data underlying the findings in their manuscript fully available?

Reviewer #1: No

Reviewer #2: No

4. Is the manuscript presented in an intelligible fashion and written in standard English?

Reviewer #1: Yes

Reviewer #2: Yes

5. Review Comments to the Author

Reviewer #1: Summary:

The paper explores a novel prospective study named the 3S-YP study. It gathers information from adolescents and young adults who have had mental health issues requiring secondary mental healthcare, and there is also a 6-month follow up. The study is novel since prior research is mainly limited to cross-sectional survey data; prior studies demonstrate associations between social media and smartphone use, but lack insight into the potential underlying mechanisms. More broadly speaking, the study aims to gather data from electronic health reports and user-generated data with the purpose of collecting objective measurements. Furthermore, it will employ natural language processing (NLP) techniques to evaluate the prevalence of self-harm.

Overall this is a well-written paper and deserves to be published; there are a few mostly minor comments which might be worth addressing. My only major comment is: If I’m not mistaken, recruitment was based on a database of people who had consented to be contacted for research. This suggests they have already been in research before or have been approached. This might be a slightly different population than a general clinical population; please discuss.

Introduction:

The introduction presents a convincing argument for the necessity of conducting a prospective study with the goal of uncovering potential underpinnings that link self-harm in youth with the usage of social media and smartphones. Firstly, there exists an alarming prevalence of self-harm among the youth, underscoring the urgency and significance of investigating its associations with digital media usage. The authors argue that existing literature is insufficient to comprehend the issue comprehensively. A key contention arises from the absence of prior prospective studies. Previous investigations on this topic predominantly adopted cross-sectional approaches, lacking the capability to provide a holistic picture or scrutinise potential underpinnings, patterns, or risk factors for self-harm in a clinically young population. Additionally, the authors posit that prior studies have not adequately considered various variables impacting mental health. Importantly, their final argument regarding current literature pertains to methodologies that heavily rely on surveys.

Some minor recommendations:

The authors mentioned that they have an "enriched" cohort. It would be helpful to explain what demographic information was included in previous studies, which in turn makes this cohort enriched.

“Although there is marked variation by age, even much younger children are engaged 80 with social media and smartphone ownership tends ” - it would be helpful to define age range

“In computational studies of users who have donated their social media data, natural language processing has shown that there are quantifiable signals present in the language used in social media postings that can identify users who are more likely to attempt suicide, with relatively high precision (16)” - High precision measured how? And were these studies cross-sectional or predictive? In addition, We might be missing some citations as it mentions “studies”.

Methods:

The 3S-YP study is a prospective observational cohort study design, conducted at South London and Maudsley NHS Foundation Trust (SLaM). Recruitment period is from June 3rd, 2021, to November 30th, 2022. The study recruited people aged 13 to 25, targeting a population served by SLaM. Utilising the Consent for Contact (C4C) patient research participation register, potential participants are identified. Eligibility criteria are applied, and researchers extract contact details from electronic health records (EHR). Recruitment is first approached via text invitations and if unsuccessful other approaches are included like, telephone calls, and email communications. Consent and enrollment are given through an online system. The study involves continuous metadata collection through an app, monthly questionnaires, and optional uploads of social media data. Data extraction from electronic health records occurs at baseline and month 6. A subset of participants agreed to be contacted for an informal phone-interview post-study. The participation in the interview was incentivized by shopping vouchers.

Study design and setting:

There is a citation error: “For full details on the study protocol, see Bye et al. (2022) (20).” It should be Bye et al. (2023).

It may be useful to provide examples about the types of services that SLaM provides.

Cohort recruitment and procedure:

“Following baseline completion, participants received automated reminders on the first and seventh day of each month for the next six months inviting them to complete the monthly questionnaires, in accordance with the schedule presented in S1 Table. To reduce participant burden, participants skipped the first monthly questionnaire in the schedule if they completed the baseline questionnaire within seven days of the following month. ” If some participants skipped the first month assessment, while other participants due to completing their baseline assessment didn’t skip that first month assessment, some participants completed one extra assessment? It would be helpful to have some clarification on what potential effect on results this might have.

“To maximise data completion at month 6, participants received additional reminders including telephone calls (in line with the standard operating procedures) and they could complete the final questionnaire at any time during the data collection period” - Clarification needed- what does it mean they could complete it at any time during data collection? How divergent from 6 months was this allowed to be? Also, missing space between “reminders” and “including”.

The authors mention standard operating procedures. It may be useful for the reader to have some understanding of what these SOPs are; perhaps in the supplementary material.

Measures:

The methods section includes an explanation for each measurement used to assess participants over a six-month period. Several outcomes of interest are outlined, beginning with primary outcomes aimed at identifying the presence of self-harm. The Child and Adolescent Self-harm in Europe (CASE) Study criteria are utilised to evaluate self-reported self-harm. Clinician-recorded self-harm history is identified through manual inspection of risk assessment forms, while current self-harm is tracked using the CRIS system. The authors include two definitions of self-harm: clinical (self-poisoning, self-injury) and a broader definition, encompassing behaviours such as disordered eating and drug misuse. Secondary outcomes include the evaluation of symptoms of anxiety, depression, sleep disturbance, and loneliness, as well as bullying victimisation, including cyberbullying. To assess social media use and smartphone use, the authors include exposure assessments following the Smartphone Addiction Scale-Short Version. Problematic smartphone use is measured by social media uploads and metadata. A more detailed explanation of what they include can be found in the protocol. All uploads and metadata were collected from consenting participants, and it was not a requirement to participate. Finally, there is a demographic section and exposure to Covid-19 data, both collected via self-reports and electronic health records. Some participants were selected to participate in an informative phone interview to capture their experience and gain feedback on their participation.

Primary outcome (self-harm):

In the protocol, there is no mention of using a broader definition of self-harm, so it would be helpful to provide a motivation regarding this inclusion. To be clear, it is a good idea but it would be helpful to motivate it and clarify. For example, if the broader definition of self-harm includes disordered eating behaviours, would a participant with an eating disorder be considered to be inflicting self-harm? Or would this ONLY be self harm IF the stated intent was self-harm? I think this is what the authors are saying but it is not 100% clear. And what if an acute substance use event that was not poisoning was experienced as self harm? E.g. a patient with BPD who takes speed after a long period of sobriety in order to harm themselves?

Secondary outcomes (symptoms/diagnosis):

The cutoff points for defining sleep disturbance differ between the two age groups. It would be helpful to clarify why this was done.

Patient and public involvement (PPI):

It would be beneficial to provide a bit more context or detail about the specific roles of SB-F and the youth mental health charity YoungMinds in the participatory process.

Results:

Results are presented in distinct sections covering recruitment, sociodemographic characteristics, self-reported and clinician-recorded history of self-harm, psychopathology, sleep disturbance, bullying victimisation and loneliness, social media and smartphone use, EHR data and data availability. The 3S-YP study, screened 1,543 young people, with 388 (30.1%) enrolling in the cohort of 362 participants. The cohort resembled the approached population, however, had a notably higher proportion of female participants (70.2%). Demographically, participants were diverse, and most were aged 18 or older (67.4%). Self-harm history was shown to be high, reported by over 80% of the cohort, with the first episode at a mean age of 12.5 (SD 3.1) and the last at 18.5 (SD 2.9) years. Participants showed high rates of anxiety (67.4%) and depression (70.4%), with approximately 40% experiencing moderate/severe sleep disturbance. Concerning social media and smartphone use, the majority were users (95.9%), mainly on Instagram (85.6%). Problematic smartphone use was observed in 48.9% of participants. High consent rates with 78.2% and 86.2% agreeing to share social media and smartphone data, respectively. Metadata from smartphone usage was available for 69.6% of Android users only. Data availability ranged from 58.9% to 100%. Sixteen participants engaged in final interviews, providing qualitative insights on overall experience.

Are mental health diagnoses available? That would be helpful to note in Table 1.

Table 2: Self-reported and clinician-recorded history of self-harm at the baseline assessment:

To provide a better picture of what kind of self-harm is present, if information is available, it could be interesting to show the frequency of different types of self-harm.

Table 5: Electronic health records data at the baseline assessment:

It would be helpful to include also a range for "Number of years since first accepted SLaM referral," given the participants' age range from 13 to 25 years old.

Age range:

Exploring the participant age range of 13 to 25 may show noteworthy differences in self-harm behaviors, symptoms, and social media/smartphone usage. It is recommended to analyze or categorize the results based on age-appropriate ranges, as comparing a 13-year-old to a 25-year-old presents challenges, for example in contexts like alcohol use, where consumption would be expected to vary between the two age groups.

Table 4: Self-reported social media and smartphone use at the baseline assessment:

Analysing social media and smartphone use with age-specific categories would provide a more accurate understanding of problematic usage. For instance, considering age-related differences in sleep patterns, such as staying up late, could offer insights into usage variations between teenagers and people in their 20s.

Discussion:

I would advise caution in calling EHRs “objective” sources of information. As a clinician who generates these records, they are far from objective and often incomplete.

If I’m not mistaken, recruitment was based on a database of people who had consented to be contacted for research. This suggests they have already been in research before or have been approached. This might be a slightly different population than a general clinical population; please discuss.

Overall:

Checking grammar and restructuring long phrases would provide better clarity throughout the paper

Citations ? missing: “Few other studies have attempted to integrate the two (42), and those that have are limited to hospital presentations for self-harm, which may not capture the broader spectrum of self-harm behaviours occurring in this population. ”

Citation missing: “Studies on youth self-harm tend to rely on either self-report measures or EHR data, with self-report subject to recall and introspective bias and EHR data dependent on access to services and data quality relevant to the research question.”

Reviewer #2: I thank the authors and editors for the opportunity to review this manuscript, which describes the baseline cohort characteristics of the 3S-YP study – a prospective, observational cohort study of young people in contact with mental health services. This cohort profile manuscript is straightforward and well-described. I only have a few minor comments:

-Make sure to spell out acronyms before their initial use. Also, it would be useful if acronyms were spelled out in Table footnotes where applicable.

-For readers not acquainted with the British system, it would be useful if more description was given about what it means to be subject to a section under the Mental Health Act.

- A portion of the “Data availability” section appears to be duplicated.

6. PLOS authors have the option to publish the peer review history of their article (what does this mean?). If published, this will include your full peer review and any attached files.

Reviewer #1: No

Reviewer #2: No

---

## [Author Response · Author response to Decision Letter 0]

22 Jan 2024

Response to Reviewer 1:

The paper explores a novel prospective study named the 3S-YP study. It gathers information from adolescents and young adults who have had mental health issues requiring secondary mental healthcare, and there is also a 6-month follow up. The study is novel since prior research is mainly limited to cross-sectional survey data; prior studies demonstrate associations between social media and smartphone use, but lack insight into the potential underlying mechanisms. More broadly speaking, the study aims to gather data from electronic health reports and user-generated data with the purpose of collecting objective measurements. Furthermore, it will employ natural language processing (NLP) techniques to evaluate the prevalence of self-harm.

Overall this is a well-written paper and deserves to be published; there are a few mostly minor comments which might be worth addressing. My only major comment is: If I’m not mistaken, recruitment was based on a database of people who had consented to be contacted for research. This suggests they have already been in research before or have been approached. This might be a slightly different population than a general clinical population; please discuss.

Thank you very much for your positive and thoughtful feedback. We have added the sentence below to the discussion section, acknowledging the potential selection bias in recruiting from a patient research participation register.

“There may be systematic differences between the C4C population, of whom we approached for this study, and the wider SLaM patient population as individual and service-related factors may influence who is approached and who consents to join the C4C register.” (Manuscript with track changes, page 37, lines 568-571)

Introduction:

The introduction presents a convincing argument for the necessity of conducting a prospective study with the goal of uncovering potential underpinnings that link self-harm in youth with the usage of social media and smartphones. Firstly, there exists an alarming prevalence of self-harm among the youth, underscoring the urgency and significance of investigating its associations with digital media usage. The authors argue that existing literature is insufficient to comprehend the issue comprehensively. A key contention arises from the absence of prior prospective studies. Previous investigations on this topic predominantly adopted cross-sectional approaches, lacking the capability to provide a holistic picture or scrutinise potential underpinnings, patterns, or risk factors for self-harm in a clinically young population. Additionally, the authors posit that prior studies have not adequately considered various variables impacting mental health. Importantly, their final argument regarding current literature pertains to methodologies that heavily rely on surveys.

Some minor recommendations:

The authors mentioned that they have an "enriched" cohort. It would be helpful to explain what demographic information was included in previous studies, which in turn makes this cohort enriched.

Thank you for this comment. We have revised the sentence below in response.

“The evidence is mainly limited to surveys or cross-sectional studies in non-clinical populations or using publicly available social media postings about mental health from unknown users, with inconsistent findings (7,8).” (Manuscript with track changes, page 5, lines 78-80)

“Although there is marked variation by age, even much younger children are engaged 80 with social media and smartphone ownership tends ” - it would be helpful to define age range

Thank you for this comment. We have revised the sentence accordingly, please see below.

“Although there is marked variation by age, even children as young as 3 years of age use social media and rates of smartphone ownership tends to increase with the move to secondary school as children approach 11 years (5)”. (Manuscript with track changes, page 5, lines 73-75)

“In computational studies of users who have donated their social media data, natural language processing has shown that there are quantifiable signals present in the language used in social media postings that can identify users who are more likely to attempt suicide, with relatively high precision (16)” - High precision measured how? And were these studies cross-sectional or predictive? In addition, We might be missing some citations as it mentions “studies”.

Thank you for these comments. We have revised this section accordingly, please see below.

“One computational study created a combined dataset of users who had donated their social media data, along with self-report data on past suicide attempts, and users who posted publicly available content describing past suicide attempts on social media, including the date of the suicide attempt. The authors used natural language processing to demonstrate that there are quantifiable signals present in the language used in social media postings that can predict risk for a suicide attempt, with relatively high precision when validated against self-report or social media (16).” (Manuscript with track changes, page 6, lines 97-103)

Methods:

The 3S-YP study is a prospective observational cohort study design, conducted at South London and Maudsley NHS Foundation Trust (SLaM). Recruitment period is from June 3rd, 2021, to November 30th, 2022. The study recruited people aged 13 to 25, targeting a population served by SLaM. Utilising the Consent for Contact (C4C) patient research participation register, potential participants are identified. Eligibility criteria are applied, and researchers extract contact details from electronic health records (EHR). Recruitment is first approached via text invitations and if unsuccessful other approaches are included like, telephone calls, and email communications. Consent and enrollment are given through an online system. The study involves continuous metadata collection through an app, monthly questionnaires, and optional uploads of social media data. Data extraction from electronic health records occurs at baseline and month 6. A subset of participants agreed to be contacted for an informal phone-interview post-study. The participation in the interview was incentivized by shopping vouchers.

Study design and setting:

There is a citation error: “For full details on the study protocol, see Bye et al. (2022) (20).” It should be Bye et al. (2023).

Thank you, we have corrected this referencing error.

It may be useful to provide examples about the types of services that SLaM provides.

We have added further detail about the types of services that SLaM provides, please see below. 

“SLaM provides the widest range of mental health services for children and adults in the UK, including community mental health, home treatment, hospital and outpatient services. SLaM serves a local population of approximately 1.3 million, as well as providing national and specialist services.” (Manuscript with track changes, page 7, lines 127-130)

Cohort recruitment and procedure:

“Following baseline completion, participants received automated reminders on the first and seventh day of each month for the next six months inviting them to complete the monthly questionnaires, in accordance with the schedule presented in S1 Table. To reduce participant burden, participants skipped the first monthly questionnaire in the schedule if they completed the baseline questionnaire within seven days of the following month. ” If some participants skipped the first month assessment, while other participants due to completing their baseline assessment didn’t skip that first month assessment, some participants completed one extra assessment? It would be helpful to have some clarification on what potential effect on results this might have.

Thank you for this comment. We have added a sentence to the manuscript, please see below, and will consider the effect of this in our analysis of the prospective data. 

“A proportion of the sample were invited to complete fewer assessments in total compared to the rest of the sample, depending on the date they completed the baseline questionnaire. The effects of which will be considered in future analyses of the prospective data to explore possible implications for data availability and retention.” (Manuscript with track changes, page 37, lines 563-566)

“To maximise data completion at month 6, participants received additional reminders including telephone calls (in line with the standard operating procedures) and they could complete the final questionnaire at any time during the data collection period” - Clarification needed- what does it mean they could complete it at any time during data collection? How divergent from 6 months was this allowed to be? Also, missing space between “reminders” and “including”.

Thank you for these comments. Participants were able to provide their month-6 questionnaire data at any time up until the end of data collection. At present, we have included all valid questionnaire data in our summary of the data available. We are likely to exclude data for the prospective analyses where there was a significant delay in that data being provided. We have included a sentence outlining this and corrected the sentence with the omitted punctuation (to note, reminders were not always including a telephone call, so we opted not to use the word ‘and’), please see below.

“To maximise data completion at month 6, participants received additional reminders, including telephone calls (in line with the standard operating procedures), and they could complete the final questionnaire at any time during the data collection period. At present, we have included all valid questionnaire data in our summary of the data available. We will consider how to handle questionnaire data for the prospective analyses where there was a significant delay in that data being provided.” (Manuscript with track changes, page 9, lines 187-193)

The authors mention standard operating procedures. It may be useful for the reader to have some understanding of what these SOPs are; perhaps in the supplementary material.

Thank you for this comment. We have added a sentence outlining what is covered in our internal SOPs, please see below.

“From our previous research and work with youth experts by experience, we have developed detailed standard operating procedures for screening and approaching potential participants, data collection, and managing risk.” (Manuscript with track changes, page 9, 169-171)

Measures:

The methods section includes an explanation for each measurement used to assess participants over a six-month period. Several outcomes of interest are outlined, beginning with primary outcomes aimed at identifying the presence of self-harm. The Child and Adolescent Self-harm in Europe (CASE) Study criteria are utilised to evaluate self-reported self-harm. Clinician-recorded self-harm history is identified through manual inspection of risk assessment forms, while current self-harm is tracked using the CRIS system. The authors include two definitions of self-harm: clinical (self-poisoning, self-injury) and a broader definition, encompassing behaviours such as disordered eating and drug misuse. Secondary outcomes include the evaluation of symptoms of anxiety, depression, sleep disturbance, and loneliness, as well as bullying victimisation, including cyberbullying. To assess social media use and smartphone use, the authors include exposure assessments following the Smartphone Addiction Scale-Short Version. Problematic smartphone use is measured by social media uploads and metadata. A more detailed explanation of what they include can be found in the protocol. All uploads and metadata were collected from consenting participants, and it was not a requirement to participate. Finally, there is a demographic section and exposure to Covid-19 data, both collected via self-reports and electronic health records. Some participants were selected to participate in an informative phone interview to capture their experience and gain feedback on their participation.

Primary outcome (self-harm):

In the protocol, there is no mention of using a broader definition of self-harm, so it would be helpful to provide a motivation regarding this inclusion. To be clear, it is a good idea but it would be helpful to motivate it and clarify. For example, if the broader definition of self-harm includes disordered eating behaviours, would a participant with an eating disorder be considered to be inflicting self-harm? Or would this ONLY be self harm IF the stated intent was self-harm? I think this is what the authors are saying but it is not 100% clear. And what if an acute substance use event that was not poisoning was experienced as self harm? E.g. a patient with BPD who takes speed after a long period of sobriety in order to harm themselves?

Thank you for these comments. To clarify, the additional behaviours included under the broader definition of self-harm were only included if there was a stated intention to self-harm. In the example described of ‘a patient with BPD who takes speed after a long period of sobriety in order to harm themselves’, this would likely be considered an act of self-poisoning as it is a discrete event involving the consumption of a recreational drug with the stated intention to self-harm, however we would explore other information available before making a clinical decision. We have added a sentence outlining the rationale for including a broader definition of self-harm and added further detail to the definitions of self-harm, please see below. 

“Self-poisoning included any events where an individual consumed more than the recommended dose of a non-recreational drug, a poisonous amount of a recreational drug if it was a discrete event, or a substance not intended for human consumption, with the intent to self-harm identified by the individual or a clinician.” (S2 File with Track Changes, Page 3)

“The decision to include this broader definition was motivated by a desire to capture events described as self-harm that would otherwise have been omitted had we solely employed a clinical definition.” (Manuscript with Track Changes, Page 13, 259-261)

“Broadly defined self-harm similarly comprised of (1) self-poisoning, (2) self-injury, and (3) both self-poisoning and self-injury, however (4) other types of self-harm also included any events involving alcohol poisoning, descriptions of other prolonged substance misuse (as opposed to a discrete self-poisoning event), and disordered eating behaviours (e.g. fasting, excessive exercise) if there was a stated intention to self-harm. We did not include any of these additional behaviours if there was not a stated intent to self-harm.” (Manuscript with track changes, page 13, lines 261-267)

Secondary outcomes (symptoms/diagnosis):

The cutoff points for defining sleep disturbance differ between the two age groups. It would be helpful to clarify why this was done.

Thank you for this comment. The different cut-offs used are in accordance with the PROMIS scoring guidelines. 

Patient and public involvement (PPI):

It would be beneficial to provide a bit more context or detail about the specific roles of SB-F and the youth mental health charity YoungMinds in the participatory process.

Thank you for this comment. We have added further detail to the manuscript outlining roles and contributions, please see below.

“We have employed a participatory research approach to promote engagement, inclusiveness and representation in this study. This has been facilitated by co-author/co-investigator and senior service user consultant – SB-F, and our charity partner, leading national UK youth mental health charity – YoungMinds,. SB-F has been a core member of the project group and instrumental in shaping the study from the outset, including contributing to research priority setting, the ethical approval process, study procedures and participant-facing materials including testing the 3S-YP app, and attending Project Steering Group meetings. YoungMinds have facilitated wider engagements through their national youth advisory programme, which has provided further opportunities to work with young people throu

---

## [Decision Letter · Decision Letter 1]

5 Feb 2024

Cohort profile: The Social media, Smartphone use and Self-harm in Young People (3S-YP) study – a prospective, observational cohort study of young people in contact with mental health services

PONE-D-23-38299R1

Dear Dr. Bye,

We’re pleased to inform you that your manuscript has been judged scientifically suitable for publication and will be formally accepted for publication once it meets all outstanding technical requirements.

Kind regards,

Vincenzo De Luca

Academic Editor

PLOS ONE

Additional Editor Comments (optional):

Reviewers' comments:

Reviewer's Responses to Questions

**Comments to the Author**

1. If the authors have adequately addressed your comments raised in a previous round of review and you feel that this manuscript is now acceptable for publication, you may indicate that here to bypass the “Comments to the Author” section, enter your conflict of interest statement in the “Confidential to Editor” section, and submit your "Accept" recommendation.

Reviewer #1: All comments have been addressed

Reviewer #2: All comments have been addressed

2. Is the manuscript technically sound, and do the data support the conclusions?

Reviewer #1: Yes

Reviewer #2: Yes

3. Has the statistical analysis been performed appropriately and rigorously? 

Reviewer #1: Yes

Reviewer #2: Yes

4. Have the authors made all data underlying the findings in their manuscript fully available?

Reviewer #1: No

Reviewer #2: No

5. Is the manuscript presented in an intelligible fashion and written in standard English?

Reviewer #1: Yes

Reviewer #2: Yes

6. Review Comments to the Author

Reviewer #1: (No Response)

Reviewer #2: I thank the authors for thoroughly addressing reviewer comments. I have no further comments at this time.

7. PLOS authors have the option to publish the peer review history of their article (what does this mean?). If published, this will include your full peer review and any attached files.

Reviewer #1: No

Reviewer #2: No

---

## [Editor Report · Acceptance letter]

26 Apr 2024

PONE-D-23-38299R1 

PLOS ONE

Dear Dr. Bye, 

I'm pleased to inform you that your manuscript has been deemed suitable for publication in PLOS ONE. Congratulations! Your manuscript is now being handed over to our production team.

Kind regards, 

on behalf of

Dr. Vincenzo De Luca 

Academic Editor

PLOS ONE